# A GREEDY APPROACH TO MAX-SLICED WASSERSTEIN GANS

## ABSTRACT

Generative Adversarial Networks have made data generation possible in various use cases, but in case of complex, high-dimensional distributions it can be difficult to train them, because of convergence problems and the appearance of mode collapse. Sliced Wasserstein GANs and especially the application of the Max-Sliced Wasserstein distance made it possible to approximate Wasserstein distance during training in an efficient and stable way and helped ease convergence problems of these architectures.

This method transforms sample assignment and distance calculation into sorting the one-dimensional projection of the samples, which results a sufficient approximation of the high-dimensional Wasserstein distance.

In this paper we will demonstrate that the approximation of the Wasserstein distance by sorting the samples is not always the optimal approach and the greedy assignment of the real and fake samples can result faster convergence and better approximation of the original distribution.

## 1 INTRODUCTION

Generative Adversarial Networks (GANs) were first introduced in Goodfellow et al. (2014), where instead of the application of a mathematically well-established loss function an other differentiable neural network, a discriminator was applied to approximate the distance between two distributions. These methods are popularly applied in data generation and has significantly improved the modelling capabilities of neural networks. It was demonstrated in various use cases that these approaches can approximate complex high-dimensional distributions in practice Karras et al. (2017), Yu et al. (2017), Brock et al. (2018).

Apart from the theoretical advantage of GANs and applying a discriminator network instead of a distance metric (e.g. $\ell 1$ or $\ell 2$ loss), modelling high-dimensional distributions with GANs often proves to be problematic in practice. The two most common problems are mode collapse, where the generator gets stuck in a state where only a small portion of the whole distribution is modeled and convergence problems, where either the generator or the discriminator solves his task almost perfectly, providing low or no gradients for training for the other network.

Convergence problems were improved, by introducing the Wasserstein distance Gulrajani et al. (2017) Arjovsky et al. (2017), which instead of a point-wise distance calculation (e.g. cross-entropy or $\ell 1$ distance) calculates a minimal transportation distance (earth mover's distance) between the two distributions.

The approximation and calculation of the Wasserstein distance is complex and difficult in high-dimensions, since in case of a large sample size calculation and minimization of the transport becomes exponentially complex, also distance can have various magnitudes in the different dimensions.

In Deshpande et al. (2018) it was demonstrated that high-dimensional distributions can be approximated by using a high number of one dimensional projections. For a selected projection the minimal transport between the one dimensional samples can be calculated by sorting both the real and the fake samples and assigning them according to their sorted indices correspondingly. As an additional advantage, it was also demonstrated in Deshpande et al. (2018) that instead of the regular mini-max game of adversarial training, the distribution of the real samples could be approximated directly by

the generator only, omitting the discriminator and turning training into a simple and more stable minimization problem. The theory of this novel method is well described and it was demonstrated that it works in practice, but unfortunately for complex, high-dimensional distributions a large amount of projections are needed.

In Deshpande et al. (2019) it was demonstrated how the high number of random projections could be substituted by a single continuously optimized plane. The parameters of this projection are optimized in an adversarial manner selecting the "worst" projection, which maximizes separation between the real and fake samples using a surrogate function. This modification brought the regular adversarial training back and created a mini-max game again, where the generator creates samples which resemble well to the original distribution according to the selected plane and the discriminator tries to find a projection, which separates the real and fake samples from each other.

The essence of Sliced Wasserstein distances is how they provide a method to calculate minimal transportation between the projected samples in one-dimension with ease, which approximates the Wasserstein distance in the original high-dimension. In theory this approach is sound Nadjahi et al. (2019) and works well in practise. It was proven in Kolouri et al. (2019) that the sliced Wasserstein distance satisfies the properties of non-negativity, identity of indiscernibles, symmetry, and triangle inequality, this way forming a true metric. However it approximates high-dimensional distributions well, we would like to demonstrate in this paper that the assignment of real and fake samples by sorting them in one dimension also has its flaws and a greedy assignment approach can perform better on commonly applied datasets. We would also argue regarding the application of the Wasserstein distance. We will demonstrate that in many cases various assignments can result the same minimal transportation during training and calculation of the Wasserstein distance with sorting can alter the distribution of perfectly modeled samples even when only a single sample differs from the approximated distribution.

## 2 WASSERSTEIN DISTANCE FOR COMPARING DISTRIBUTIONS

Generative adversarial networks can be described by a generator ($G$), whose task is to generate a fake distribution ($\mathbb{P}_F$), which resembles a distribution of real samples ($\mathbb{P}_R$) and a discriminator, whose task is to distinguish between $\mathbb{P}_F$ and $\mathbb{P}_R$ samples. Formally the following min-max objective is iteratively optimized:

$$\min_G \max_D V(\mathbb{P}_F, \mathbb{P}_R) \tag{1}$$

Where $V$ is a distance measure or divergence between the two distributions. In Arjovsky & Bottou (2017) the Wasserstein-p distance was proposed to improve the staibility of GANs, which can be defined as:

$$\mathbb{W}_\mathbb{P}(\mathbb{P}_F, \mathbb{P}_R) = \inf_{\gamma \in \prod(\mathbb{P}_F, \mathbb{P}_R)} (\mathbb{E}_{(x,y) \rightsquigarrow \gamma} \|x - y\|^p)^{\frac{1}{p}} \tag{2}$$

where $p$ is a parameter of the distance, and $\prod(\mathbb{P}_F, \mathbb{P}_R)$ defines all the possible joint distributions over $\mathbb{P}_F$ and $\mathbb{P}_R$. The number of possible joint distributions increases factorially with the number of samples and the calculation of the minimal transport can be difficult.

The instability and high complexity of wasserstein GANs were further improved by the introduction of the sliced Wassersstein distance in Deshpande et al. (2018), which can be defined as:

$$\mathbb{W}_\mathbb{P}(\mathbb{P}_F, \mathbb{P}_R) = \left[ \int_{w \in \Omega} W_p^p(\mathbb{P}_F^w, \mathbb{P}_R^w) dw \right]^p \tag{3}$$

where $\mathbb{P}_F^w, \mathbb{P}_R^w$ are one-dimensional projections of the high dimensional real and fake samples to w and $\Omega$ denotes a sufficiently high number of projections on the unit sphere. In this setting the Wasserstein distance can be calculated by sorting both the real and fake samples and assigning them by their indices, instead of checking all the possible joint distributions in $\prod(\mathbb{P}_F^w, \mathbb{P}_R^w)$.

Max-sliced Wasserstein GANs can be introduced by selecting the worst projected distribution from $\Omega$ noted by $w_{max}$ and since the projection can be implemented by a fully connected layer, one can re-introduce the mini-max game used in regular GAN training:

$$\min_G \max_{D(w \in \Omega)} \mathbb{W}_\mathbb{P}(\mathbb{P}_F, \mathbb{P}_R) \tag{4}$$

In this setup the distance in the single, one-dimensional projected space can be calculated by sorting the samples in the similar manner as in case of the sliced Wasserstein distance.

## 2.1 A Criticism of The Wasserstein Distance

In case of high sample number and complex high-dimensional distributions the optimal transport can often be calculated using various assignments between two sets of samples. For example if a sample containing $n$ elements from $\mathbb{P}_F$ can be defined by the series of $F^w_{1...n}$ after the projection, and a sample from $\mathbb{P}_R$ is defined by $R^w_{1...n}$ and we know that for every $i$ and $j$ ($i, j \in 1...n$):

$$F^w_i < R^w_j \tag{5}$$

Which means that the projected samples are well separated from each other (all the projected fake samples are smaller than any of the projected real samples), for $p = 1$ all possible assignments of the $i, j$ pairs will result the same distance, which is the minimal transportation of the samples, although the pairwise differences can be very different.

One can also easily see, that the minimal transport calculated with sorting might not assign identical samples to each other, which is depicted on Fig. 1. As it can be seen from the figure, for example in case of two Gaussian distributions with the same variance, but different mean values using sorting to calculate the minimal transport (along arbitrary dimensions), will results the shift of one of the distributions (in this case green). This transformation will also affect those samples which are at the intersection of the two distributions. This means that there might be identical pairs in ($\mathbb{P}_F$ and $\mathbb{P}_R$) generating an error, which is not correct. One could assume that if a generator produces a sample identical to one of the samples of $\mathbb{P}_R$ it should not generate an error and no gradients should be invoked by this sample pair. In this case the assignment will not pair these identical samples for comparison.

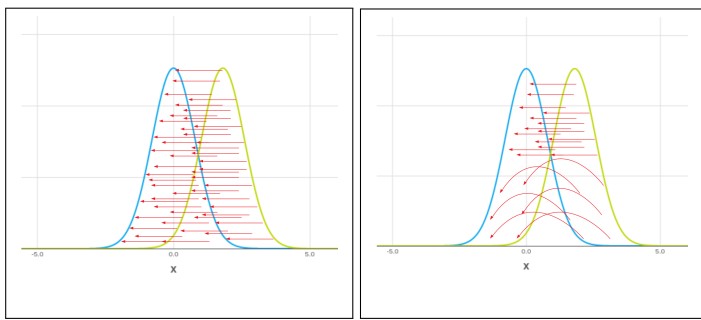

Figure 1: This figure depicts a flaw of the Wasserstein distance. In these figures a blue (desired output distribution and green, generated fake distribution can be seen), the assignments/desired transports are illustrated by red arrows. Wasserstein distance calculates minimal transportation between all the samples. In many cases this results shifting all the samples to cover the whole distribution correctly, this is depicted on the left subfigure. Unfortunately this means that those samples which are at the intersection of the two distributions, meaning that for these generated samples an identical pair could be found in the real samples, will also be altered and shifted towards an other part of the real distribution. Instead of this we propose the calculation of the assignment using a greedy method. This will ensure that that identical (or similar) sample pairs will be selected first, and after this the transportation between the disjunct regions will be calculated, which is depicted on the right subfigure.

Instead of sorting the samples we propose a method which assigns similar samples to each other. First we would like to remove the intersection of the two distributions and calculate the minimal transport for the remaining disjunct regions. By this calculation the value of the minimal transport will be the same (in case of a sufficiently large sample number), but more similar samples will be assigned to each other.

The previous example reveals a caveat of the Wasserstein distance, that it optimizes the global transport between two distributions, but ignores identical sample pairs or intersecting regions of density

functions in them. Unfortunately identical samples will be extremely rarely found in two distributions, but closer sample pairs should also be assigned to each other. In training one typically uses mini-batches and will not see the whole distribution, certain parts or elements might not be present in each mini-batch if they probability in the original distribution is lower than one over the mini-batch size. This can cause further problems in the calculation of the minimal transport using sorting, since the appearance of one wrongly generated sample at the first or last position of the projected samples can result a completely different assignment. This problem is depicted on Fig. 2. We have to emphasize that although the assignment and distance calculation happen in one dimension, these distances are results of the projection of high dimensional embeddings, where there can exist dimensions which can decrease the distance between the two samples without significantly changing the projected position of all the other samples.

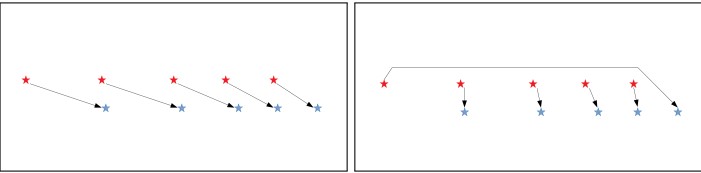

Figure 2: In this figure we would like to demonstrate different approaches for minimal transportation in one-dimension using mini-batches. Real samples are noted by blue and fake samples are plotted by red stars. The position of the samples were shifted vertically for better display, but only their horizontal position matters. On both figures the same constellation can be seen: four similar sample pairs and one which is very different. The left subfigure depicts sample assignment by sorting, meanwhile the right figure depicts assignment by a greedy approach. The summed distances are the same in both cases for $p = 1$. We would like to argue that the lower assignment can be better for network training.

## 2.2 A GREEDY APPROACH FOR SAMPLE ASSIGNMENT

We introduce greedy training of max sliced Wasserstein GANs, where instead of sorting the samples of the projections we iteratively select the most similar pairs of them for loss calculation. Our training algorithm as a pseudocode is defined in Algorithm 1. The training process is really similar to the approach introduced in Deshpande et al. (2019). The main and only alteration is between line ten and sixteen, where instead of the sorting operation of the original approach we first generate a matrix which determines the distances for all possible sample pairs in one-dimension. First we select the smallest element in this matrix and remove its row and column and iterate this process until the matrix contains only one element which will be the distance between the last two, least similar sample pairs.

We have to note that our approach requires $O(n^3)$ steps compared to the original $O(nlog(n))$ operations of sorting. We have to implement $n$ minimum selections in the distance matrix with size $n \times n$. We also have to note that this increase in complexity refers for training only and has no effect ton the inference complexity of the generator, also $n$ is the sample or mini-batch size during training, which is usually a relatively low number. In our experiences, using batches of 16 to 512 samples, this step did not result a significant increase in computation time during training.

## 2.3 HYBRID APPROACH FOR SAMPLE ASSIGNMENT

In the previous section flaws of the Wasserstein distance using sorting in one-dimension were presented and greedy sample assignment was suggested as a possible method fixing these defects. On the other hand one has to admit that in certain cases greedy approach does not ensure minimal transportation, transportation resulted by sorting will always be lower or equal to the assignment calculated by the greedy approach. We also have to state that distance calculation with the greedy assignment does not form a proper metric since it does not satisfy the triangle inequality. It can also easily be seen that one could generate one-dimensional cases where the greedy assignment is arbitrarily larger than sorting. To ensure that these cases can not occur during training we have introduced a hybrid approach where we first sort the samples and calculate their Wasserstein distance (noted as $\mathbb{W}_{\mathbb{P}}^S$) and after this step we also calculate the distance using greedy assignment (noted as

---

**Algorithm 1** Training the Greedy Max-Sliced Wasserstein Generator

---

**Given:** Generator parameters $\theta_g$, discriminator parameters $\theta_d$, $\omega_d$, sample size $n$, learning rate $\alpha$

1: **while** $\theta_g$ not converged **do**
2:     **for** $i = 1 \rightarrow n$ **do**
3:         Sample data $\{\mathbb{D}^i\}_{i=1}^n \backsim \mathbb{P}_R$ generated samples $\{\mathbb{F}_{\theta_g}^i\}_{i=1}^n \backsim \mathbb{P}_F$
4:         compute surrogate loss $s(\omega\mathbb{D}^i, \omega\mathbb{F}_{\theta_g}^i)$
5:         **return** L $\leftarrow$s$(\omega\mathbb{D}^i, \omega\mathbb{F}_{\theta_g}^i)$
6:         $(\hat{\omega}, \hat{\theta}_d) \leftarrow (\hat{\omega}, \hat{\theta}_d) - \alpha\nabla_{\hat{\omega}, \hat{\theta}_d} L$
7:     **end for**
8:     compute max-sliced Wasserstein Distance $max - W_p(\omega\mathbb{D}^i, \omega\mathbb{F}_{\theta_g}^i)$
9:     Sample data $\{\mathbb{D}^i\}_{i=1}^n \backsim \mathbb{P}_R$ generated samples $\{\mathbb{F}_{\theta_g}^i\}_{i=1}^n \backsim \mathbb{P}_F$
10:    $L \leftarrow 0$
11:    Create distance matrix: $M_{kl} = \left\| \omega\mathbb{D}_k^i - \omega\mathbb{F}_{\theta_g}^i l \right\|$
12:    **for** $k = 1 \rightarrow n$ **do**
13:        Find min value of $M$: $min(M) = m_{s,t}$
14:        $L \leftarrow L + m_{s,t}$
15:        Remove Row $s$ and Column $t$ from $M$
16:    **end for**
17:    $\theta_g \leftarrow \theta_g - \alpha\nabla_{\theta_g} L$
18: **end while**

---

$\mathbb{W}_{\mathbb{P}}^G$). In this case a parameter ($\nu$) can be set, determining a limit and if the difference between the two distances is larger than this value the sorted distance will be used. This way Wasserstein distance with the hybrid assignment ($\mathbb{W}_{\mathbb{P}}^H$) can be defined as:

$$\mathbb{W}_{\mathbb{P}}^H = \begin{Bmatrix} \mathbb{W}_{\mathbb{P}}^S & , if : (\mathbb{W}_{\mathbb{P}}^G - \mathbb{W}_{\mathbb{P}}^S) > \nu\mathbb{W}_{\mathbb{P}}^S \\ \mathbb{W}_{\mathbb{P}}^G & otherwise \end{Bmatrix} \tag{6}$$

In case of $\nu = 0$ the greedy assignment will only be used in those cases, where it also results the minimal transportation. If $\nu$ is set to one, the greedy assignment will be used if the distance calculated this way is less than twice the minimal distance. We were using $\nu = 1$ in our experiments. It is also worth to note, that sorting the samples before creating the distance matrix ($M$) can help finding the minimal element, since in this case all series starting from the minimal element in a selected row or column will form a monotonically increasing series.

## 3 COMPARISON OF THE DIFFERENT APPROACHES AND RESULTS

### 3.1 ONE-DIMENSIONAL GAUSSIAN MIXTURE MODEL

For the first tests we have generated a one-dimensional toy problem using Gaussian Mixture Model with five modes, each of them with variance of 0.15 and expected values of 0, 2, 4, 6 and 8 accordingly. We have trained a four layered fully connected network containing 128, 256, 512 and 1 neurons in the corresponding layers for 5000 iterations using Adam optimizer, with learning rate of 0.001, with batches of 64. No discriminator was applied in this case, the position of the generated samples were optimized directly by the generator and distance calculation was done directly on the output of the generator. In one setup the loss was calculated by sorting the samples, in the other setup we used the greedy approach introduced in Algorithm 1. For further details regarding the hyperparameters of all simulations and for the sake of reproducibility our code can be found on github, containing all details of the described simulations: *the link was removed from the text but for the sake of anonymity, but a link pointing to the codes was used during submission*

After training we have used the generator to generate 16000 samples and we have compared it to 16000 samples generated by the Gaussian Mixtures Model. We have calculated the Kullback-Leibler divergence and the Pearson correlation coefficient between these two distributions, repeated

all experiments ten times and averaged the results. The results can be seen in Table 1. As it can be seen training the network with greedy assignment resulted lower Kullback-Leibler divergence and higher correlation between the generated and the original distributions, which signals the the greedy approach could approximate the original distribution of the samples better. For the calculation of the Kullback-Leibler divergence histogram calculation was done using 100 bins uniformly distributed between -0.5 and 8.5. An image plotting the histograms using these parameters can be seen on Fig. 3.

Table 1: KL divergence, and Pearson correlation between the original and generated distribution using sorting and a greedy approach.

| Method | KL DIV | Pearson Corr |
|--------|--------|--------------|
| Sorted | 0.91 | 0.41 |
| Greedy | **0.68** | **0.74** |

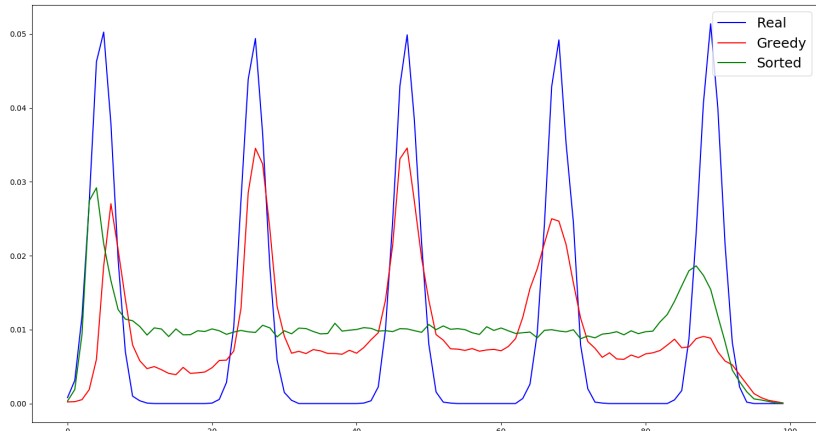

Figure 3: Histograms of the one-dimensional distributions. The blue curve plots the original distribution of the Gaussian Mixture Model, the green curve depicts the generated distribution using sorting for sample assignment, meanwhile the red curve display the result of the greedy approach.

## 3.2 TWO-DIMENSIONAL GAUSSIAN MIXTURE MODEL

We have repeated the simulations described in section 3.1 using two-dimensional Gaussian Mixture models. Nine modes were generated forming a small $3 \times 3$ grid, each with variance of 0.1 and the distance between neighbouring modes was 3.5 in the grid. In this case a five-layered network with 128, 256, 512, 1024 and 1 neurons was applied and training was executed for 500.000 iterations. All other parameters of the simulation were kept the same. The Kullback-Leibler divergence and Pearson correlation coefficients of these simulations comparing the greedy and sorting approaches can be seen in Table 2, meanwhile random samples are depicted on Fig. **??**. for qulaitative comparison. In this experiment a two-dimensional histogram was used for the Kullback-Leibler divergence calculation, where a uniform grid was formed between -2 and 8 forming 100 bins.

## 3.3 EXPERIMENTS ON THE MNIST DATASET

We have evaluated our approach on the MNIST dataset (LeCun (1998)) as well, where we have used the DCGAN architecture Radford et al. (2015) for image generation. Images were resized to $64 \times 64$ to match the input dimension of the architecture, but single channel images were used. We were using batches of 128 and 16 samples and Adam optimizer for training both the generator and the discriminator (which was a single projection). Since the comparison of high-dimensional (in this case 4096) distributions is complex, we have binarized the images (thresholded at half of the maximum intensity) and calculated the KL divergence between the distribution of white and black

Table 2: KL divergence, and Pearson correlation between the original and generated distribution using sorting and a greedy approach for the two-dimensional Gaussian Mixture Model.

| Method | KL DIV | Pearson Corr |
|--------|--------|--------------|
| Sorted | 1.21 | 0.55 |
| Greedy | **0.33** | **0.80** |

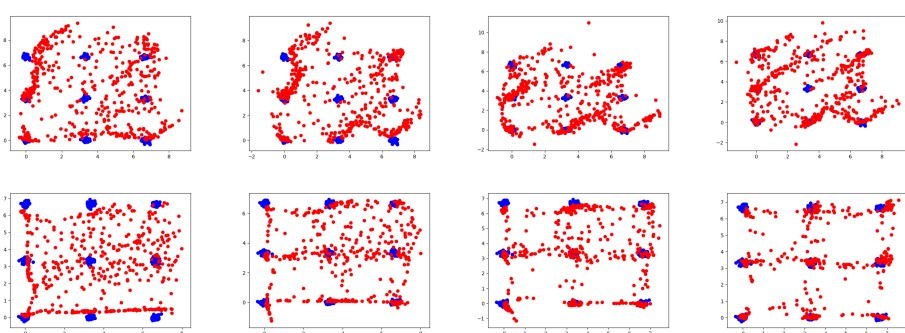

Figure 4: This Figure depicts the real (blue) and the generated (red) samples for two-dimensional Gaussian Mixture Models. The upper rows were generated using sorting, meanwhile the lower samples were produced using greedy assignment. The subfigures from left to right display results at 2, 3, 4 and five hundred thousand iterations.

values at each pixel for the original distribution and 60000 generated fake samples. The results can be seen in Table 3, meanwhile a qualitative comparison of randomly selected samples can be seen on Fig. 5.

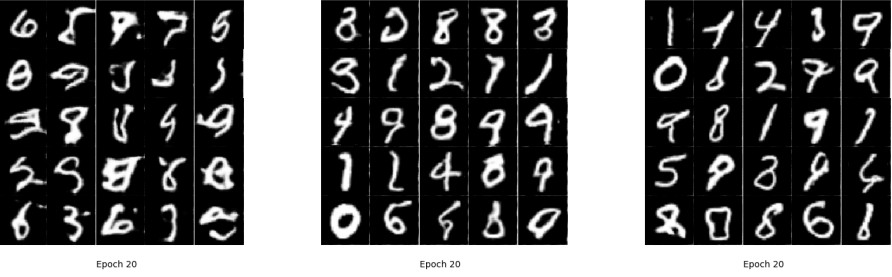

Figure 5: This Figure displays randomly selected samples generated with the same network architecture and training parameters on the MNIST dataset using batches of 16. The samples on the left were generated using sorting assignment with Max-sliced Wasserstein distance, the samples in the middle were generated using greedy sample assignment and the samples on the right were generated using the hybrid approach. All samples were generated after 20 epochs of training.

### 3.4 EXPERIMENTS ON THE CELEBA DATASET

We have also conducted experiments on the resized CelebA dataset (Liu et al. (2018)), in our case images were downscaled to $64 \times 64$. We have used the DCGAN architecture and compared three different approaches for sample assignment for distance calculation. We did not use any special normalization or initialization methods during training. We used batches of 16 for training.

We have selected 10.000 samples randomly from the dataset and generated 10.000 random projection and used sliced Wasserstein distance with sorting to compare the distributions along these

Table 3: KL divergence between the original and generated distributions using sorting, greedy and a hybrid approach and different batch sizes (16 and 128) on the MNIST dataset.

| Method | KL DIV |
|--------|--------|
| Sorted16 | 0.0927 |
| Greedy16 | 0.0233 |
| Hybrid16 | **0.0147** |
| Sorted128 | 0.0489 |
| Greedy128 | 0.0172 |
| Hybrid128 | **0.009** |

Table 4: The sliced Wasserstein distance on the generated distributions on the CelebA dataset using sorting, greedy method and a hybrid approach for sample assignment.

| Method | Sliced Wass. |
|--------|--------------|
| Sorted | $3.472 \times 10^{-3}$ |
| Greedy | $1.294 \times 10^{-3}$ |
| Hybrid | $0.783 \times 10^{-3}$ |

lines. The results can be seen in Table 4, meanwhile a qualitative comparison of randomly selected samples can be seen on Fig. 6.

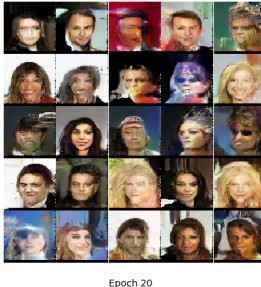 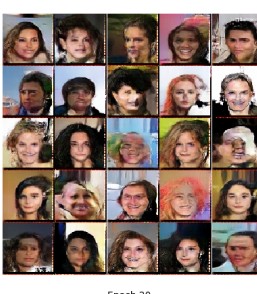 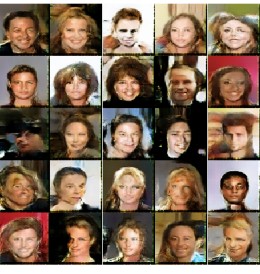

Epoch 20          Epoch 20          Epoch 20

Figure 6: This Figure depicts random samples generated with the same network architecture and training parameters on the CelebA dataset using batches of 16. All samples were generated using Max-sliced Wasserstein distance, the samples on the left were generated using sorting for sample assignment, in the middle the greedy approach was used, meanwhile the samples on the right were generated using the hybrid approach. All samples were generated after 20 epochs of training.

## 4 CONCLUSION

In this paper we have introduced greedy sample assignment for Max-Sliced Wasserstein GANs. We have shown that using one-dimensional samples, in many cases multiple assignments can result optimal transportation and in most cases sorting changes all the samples, meanwhile those parts of the distribution which are at a "good" position should not generate error.

We proposed greedy assignment as a possible solution, where samples will be assigned to their most similar counterparts. We have also introduced how the combination of the two methods can be applied resulting a hybrid approach in which it can automatically selected - based on the difference of the two measures - which assignment will be used.

We have demonstrated on simple toy datasets that greedy assignment performs better than sorting the samples and we have evaluated both the greedy and the hybrid methods on commonly investigated

datasets (MNIST and CelebA). With all datasets the greedy assignment resulted lower Kullback-Leibler divergence and higher correlation than the traditional approach.

We have used the Max-Sliced Wasserstein distance for the base of our comparison, since this is the most recent version, which also results the best performance, but all the approaches can be exploited in case of regular Sliced Wasserstein distances as well. Also our approach changes the distance calculation only and it can be applied together with various other improved techniques and architectures which are used in GAN training.

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
