# OpenReview forum: "A Greedy Approach to Max-Sliced Wasserstein GANs"
_ICLR.cc/2020/Conference — Reject_

### Official Review · AnonReviewer1 · 2019-10-22
**Official Blind Review #1**

**Rating:** 1

**Review:**

The paper proposes a variant of the max-sliced Wasserstein distance, where instead of sorting, a greedy assignment is performed. As no theory is provided, the paper is purely of experimental nature.

Considering the above, the experimental evaluation is way too preliminary:

- Looking at the generated images, much better results can be achieved with a Vanilla-GAN + GP regularization, so it is completely unclear to me why the proposed GAN should be used, as it is seems more complicated to implement.

- The KL divergence evaluation seems non-standard, and it is not explained why this metric is chosen over the standard ones (FID, Inception Score). However, I think a evaluation with respect to standard metrics is a must for an experimental GAN paper.

- I would have liked to see a comparison to using the exact Wasserstein distance, as it also scales roughly like n^3. For example, the recent paper "Wasserstein GAN with Quadratic Transport Cost" computes the exact distance using linear programming, and there it is shown to yield good results w.r.t. FID.

Minor comments I noticed during reading (no impact on my rating):

- In Eq. 1, the maximization over D is confusing given that in the following Eq. 2 the primal form of the Wasserstein distance is shown.

- The number of possible joint distributions is a continuum, and not a discrete quantity that increases factorially. Anyway if both distributions are discrete, then the problem can be reduced to a discrete optimization problem with factorially many candidate solutions, but this is very misleading as there are polynomial time algorithms.


**Experience Assessment:**

I have published one or two papers in this area.

**Review Assessment: Checking Correctness Of Derivations And Theory:**

N/A

**Review Assessment: Checking Correctness Of Experiments:**

I carefully checked the experiments.

**Review Assessment: Thoroughness In Paper Reading:**

I read the paper at least twice and used my best judgement in assessing the paper.

---

### Official Review · AnonReviewer3 · 2019-10-24
**Official Blind Review #3**

**Rating:** 1

**Review:**

The paper suggests a new way to train max slides Wasserstein GANs. I find that the paper has too few innovation in comparison with the approach introduced in Deshpande et al. (2019) (reference in the paper). The authors themselves described that the difference is ‘instead of sorting the samples of the projections we iteratively select the most similar pairs of them for loss calculation’. I think it is not enough for a publication.

Minor suggestions:
* Page 3. ‘As it can be seen from the figure, for example…’ I think ‘for example’ here is redundant.  In the same sentence ‘will results’ -> ‘will result’.
* Section 2.2. ‘First we select the smallest element…’ I would remove ‘first’ because it was used in the previous sentence.
* Section 2.2. has no effect ton-> on
* Equation (6). There must be a comma before ‘otherwise’. And maybe it would look better to write the equation as a system (with one left braсket).
* Section 3.2. There is ‘Fig. ??’

**Experience Assessment:**

I do not know much about this area.

**Review Assessment: Checking Correctness Of Derivations And Theory:**

I assessed the sensibility of the derivations and theory.

**Review Assessment: Checking Correctness Of Experiments:**

I did not assess the experiments.

**Review Assessment: Thoroughness In Paper Reading:**

I made a quick assessment of this paper.

---

### Official Review · AnonReviewer2 · 2019-10-25
**Official Blind Review #2**

**Rating:** 1

**Review:**

This paper proposes two alternative approaches to max-sliced Wasserstein GANs. They are based on the authors’ claim that there is a “flaw” in the Wasserstein-1 distance between probability distributions on the one-dimensional space. Briefly, the authors’ argument says that the “flaw” is that the optimal transport may not be unique, some of which are better for network learning than others. One proposal, described in Section 2.2, is to find a plausible transport plan in a greedy manner. The other proposal, described in Section2.3, is a hybrid of the greedy approach in Section 2.2 and the original sliced Wasserstein distance.

The working hypothesis in this paper, that the above “flaw” is indeed problematic in learning in max-sliced Wasserstein GANs, has not been confirmed in any sense in this paper. Algorithm 1 is meant to explain one of the proposal, the greedy approach, but I found that several undefined symbols are used there, so that it seems hard to understand it. Numerical experiments in Section 3 are not well described. Because of these, I would not be able to recommend acceptance of this paper.

Algorithm 1 seems to heavily rely on Algorithm 1 in Deshpande et al., 2019. This paper does not provide any explanation about why one should sample n data for each i running from 1 to n (line 3), what the “surrogate loss” is (line 4), what ¥omega is (line 4), why one should care about the surrogate loss between ith data and ith generated sample (line 4), as well as what D^i_k means (line 11).

I do not understand how the Pearson correlation coefficient between the generator (or fake) distribution and the real distribution in Section 3. As far as my understanding, fake samples and real samples are sampled independently, so that correlation coefficient should ideally vanish in any case. Also, the KL divergence is not symmetric, so that whether KL(P_F,P_R) or KL(P_R,P_F) was evaluated has to be explicitly stated. Furthermore, recalling that the Wasserstein distance has originally been introduced to the GAN literature in order to alleviate the problems associated with the KL-based divergence (Jensen-Shannon), I do not understand either why the authors chose to use the KL divergence in their performance comparison.

The “flaw” argued in this paper does not apply to the Wasserstein distance in general, but specifically to the Wasserstein-1 distance between one-dimensional distributions. This fact should be stated clearly.

Page 2, equation (2): The subscript ¥mathbb{P} should read p.
Page 2, two lines below equation (3): w should be italicized.
Page 4, line 4: if the(y->ir) probability
Page 4, line 24: has no effect (t)on the inference complexity
Page 5, Algorithm 1, line 17: g of ¥theta g should be a subscript of ¥theta.
Page 6, line 3: which signals th(e->at) the
Page 6, line 12: was executed for (500.000->500,000) iterations.
Page 6, line 15: Figure number is missing.
Page 7, lines 9-10: (10.000->10,000) random projection(s) and used


**Experience Assessment:**

I have read many papers in this area.

**Review Assessment: Checking Correctness Of Derivations And Theory:**

I assessed the sensibility of the derivations and theory.

**Review Assessment: Checking Correctness Of Experiments:**

I assessed the sensibility of the experiments.

**Review Assessment: Thoroughness In Paper Reading:**

I read the paper at least twice and used my best judgement in assessing the paper.

---

### Decision · Program_Chairs · 2019-12-19

**Decision:**

Reject

**Comment:**

The paper proposes a variant of the max-sliced Wasserstein distance, where instead of sorting, a greedy assignment is performed. As no theory is provided, the paper is purely of experimental nature.

Unfortunately the work is too preliminary to warrant publication at this time, and would need further experimental or theoretical strengthening to be of general interest to the ICLR community.